# Decomposition and Nitrogen Release Rates of Foliar Litter from Single and Mixed Agroforestry Species under Field Conditions

Magnolia del Carmen Tzec-Gamboa [1], Oscar Omar Álvarez-Rivera [2], Luis Ramírez y Avilés [1] and Francisco Javier Solorio-Sánchez [1,*]

[1] Faculty of Medicine Veterinary and Animal Sciences, University of Yucatán, Km 15.5 Carretera Mérida-Xmatkuil s/n, Mérida 97315, Mexico
[2] Biotechnology Unit, Scientific Research Center of Yucatan A.C., Calle 43 No. 130 by 32 and 34, Col. Chuburna de Hidalgo, Mérida 97205, Mexico
* Correspondence: ssolorio@uady.mx or ssolorio@correo.uady.mx

**Abstract:** Decomposition and N release pattern from the leaves of three shrubs species were studied under field conditions. Leaves of *Leucaena leucocephala* (Lam.), *Guazuma ulmifolia* (Lam.) and *Moringa oleifera* (Lam.) and two mixtures, *Leucaena + Moringa* and *Leucaena + Guazuma*, in a complete randomized block design, were studied during the dry and wet seasons. Litterbags were randomly distributed in each experimental block and placed on the soil surface, and residues were recovered after 2, 4, 8, and 16 weeks. Double exponential model decay was better fitted to describe the pattern of the decay of the release of various leaf constituents. Litter dry weight loss and N release were faster from *Moringa*, followed by the *Leucaena + Moringa* mixture, while the *Guazuma* leaf litter decomposed much slower. In the wet period, a rapid N release was observed for *Moringa* (60%) and *Leucaena + Moringa* (43%) in the first two weeks. In contrast, *Guazuma* and the *Leucaena + Guazuma* mixture released about 46% of N in 16 weeks. In the dry period, leaves released most of their N during the first 8 to 16 weeks. *Moringa* and *Leucaena + Moringa* ranked first, having lost 81 and 75% of its initial N, respectively. The ratios of condensed tannin and polyphenols to N were significantly correlated with the N released. It was concluded that the initial mass loss from the leaf litter was high and rapid in the rainy period in comparison to the dry period. The residue disappearance pattern of *Moringa*, *Leucaena* and *Leucaena + Moringa* followed an asymptotic model, with more than 80% of the original residue released during the 16-week study period.

**Keywords:** soil fertility; litter decomposition; mixed litter; nutrient cycling

## 1. Introduction

Food insecurity has increased in recent years. In 2020, the most-affected regions were Latin America and the Caribbean and Africa; in Latin America and the Caribbean the prevalence of moderate or severe food insecurity was 41%, and the prevalence of severe food insecurity was 14% [1,2]. In the Yucatan Peninsula, forests have been cleared, mainly to provide land to grow crops and raise livestock [3]; in addition, the extraction of timber, fodder, and other forest products can result in negative nutrient balances and resultant reduced soil fertility [4].

In general, the decline in soil productivity in the tropics has resulted in a search for methods that use more efficient nutrient cycling to sustain agricultural production. For example, there has been a growing emphasis in tropical countries on encouraging farmers to add organic material to agricultural land. However, increasing biological nitrogen fixation has often been proposed as the best strategy for raising productivity without further damage to natural resources [5]. In the tropics, trees are increasingly being recommended for land restoration where soil has been degraded [6], for fallow improvement and for erosion control. Examples of benefits in soil improvement are well documented [7–9]. Many

of the nutrients required for plant growth can be provided by the decomposition of tree leaf litter [10,11]. In tropical agroecosystems, the decomposition of the leaf litter and the subsequent nutrient release thus represent an important management technique by which poor farmers can reduce external inputs and maintain or increase agricultural production. Fodder trees or shrubs can provide both nutrient-rich foliage for animal feed and nutrient input to the soil via litter-fall or mulch. Long-term sustainability depends on nutrient loss from the system, e.g., from the sale of agricultural products being balanced by inputs.

In agroforestry systems, the quantity and quality of residue from tree prunings or litterfall influences soil aggregation and enhances levels of soil organic matter [12]. The potential of these resources to contribute nutrients, especially N, for other crops is highly dependent on their N release characteristics with respect to demand for uptake by the crops [13]. Litter decomposition and the release of mineral-N are influenced by environmental factors and plant chemical composition, which may be described by lignin, C, N and polyphenol content [14], which is determined largely by species and consequently their potential to supply nutrients and organic matter to the agroecosystems.

Non-legume species, such as *Guazuma ulmifolia,* tend to produce litter with low nutrient content that therefore would decompose slowly; in contrast, $N_2$-fixing species, such as *Leucaena leucocephala,* have higher N content in their biomass and decompose more rapidly [15,16]. Although there have been many studies of decomposition and nutrient release in the tropics, many of them have focused on how the litter of individual species decomposes [17,18], and rather little research has been carried out on litter from species mixtures [19–22]. The studies on litter-mixture decomposition have shown differences with respect to what might be expected based on the additive decomposition of the litter in monoculture species. Mixing leaves from species of differing leaf structure and quality can influence decomposition and decay rates through the transfer of nutrients and secondary chemicals among litter types [23]. Interactions can either enhance or slow down the decomposition process through the positive interactions likely to occur if one of the components is relatively rich in nutrients. These nutrients could aid the decomposition of the other species in the mixture and are supposedly translocated from one litter type to another [24].

Mixing *G. ulmifolia litter* with *L. leucocephala* (N rich litter) may enhance the decomposition of *G. ulmifolia* litter. In terms of mass loss, synergistic effects occurred in most mixed litter decomposition studies, ranging between 1% and 65%, whereas additive or antagonistic effects were less frequently observed to range from 1.5% and 22% [18]. Knowledge of the causes of these factors will greatly improve the probability of success.

In mixtures containing a $N_2$-fixing species and due to the nutritional interactions in mixtures, N become available to non-$N_2$-fixing plants after plant tissue decomposition, releasing N that cycles through the ecosystem [25], so more soil N may be available to non-$N_2$-fixing plants before the fixed N is cycled and transferred to the non-$N_2$-fixing plants [21]. The present experiment was designed to compare the rates of decomposition and nitrogen release from the leaf litter of (1) three fodder shrub species: *Leucaena leucocephala*, *Guazuma ulmifolia* and *Moringa oleifera,* and (2) mixtures of the $N_2$-fixing *Leucaena* with the other two species.

## 2. Materials and Methods

### 2.1. Description of the Experimental Site

The study was carried out at the research field station of the Faculty of Medicine Veterinary and Animal Sciences, University of Yucatán, Mérida, in the Yucatán peninsula, southern México. The field station lies at 21°15′ N and 90°25′ W at an altitude of 10 m a.s.l. in the sub-humid climatic zone, with a total annual rainfall of 960 mm, and a 6–7-month dry period [26]. From November to February, the mean daily temperature is 23 °C (max 32 °C, min 15 °C), while in March to October, it rises to 30 °C (max 37 °C, min 23 °C). The landscape is flat; the soils are calcareous and mainly shallow (<0.10 m in depth), with much of the surface being exposed limestone or rocky outcrops that are relatively well-drained. The soils are classified as Leptosols and are of moderate fertility with 1.0 to

1.5% organic carbon content and a pH range of 7.5 to 7.8. The soils (upper 0.10 m) of the experimental site were characterized, we determine the $pH_{H2O}$ (ratio 1:2) [27], total nitrogen (method: Kjeldahl) [28], organic carbon (Method: Walkley–Black) [29], phosphorus (method: Olsen) [30], potassium and calcium (method: flame emission spectroscopy) [31] and magnesium (method: titration) [31] (Table 1).

**Table 1.** Physical and chemical characteristics of the experimental area (mean values by block).

| Block | Stone | pH | N | C | C:N | P | Exch K | Exch Ca | Exch Mg |
|---|---|---|---|---|---|---|---|---|---|
| | % | | % | | Ratio | | mg kg$^{-1}$ | | |
| I | 78 | 7.8 | 0.89 | 6.4 | 7.2 | 28 | 530 | 872 | 352 |
| II | 60 | 7.8 | 0.98 | 5.0 | 5.1 | 45 | 565 | 824 | 328 |
| III | 79 | 7.9 | 0.99 | 7.2 | 7.3 | 81 | 457 | 1077 | 310 |
| IV | 79 | 7.9 | 0.96 | 6.1 | 6.4 | 111 | 517 | 1573 | 388 |
| Mean | 74 | 7.8 | 0.95 | 6.2 | 6.5 | 66 | 517 | 1086 | 345 |

*2.2. Field Experimental Procedure*

As part of a larger research program, a decomposition experiment was carried out using litter bags in a complete randomized block design with five treatments. The treatments were: 1. *Leucaena leucocephala*, 2. *Guazuma ulmifolia*, 3. *Moringa oleifera*, 4. a mixture of *Leucaena* + *Moringa* and 4. another mixture of *Leucaena* + *Guazuma*, *Leucaena* is a $N_2$-fixer.

A total of 160 bags was used in each experiment, i.e., 5 treatments × 4 blocks × 4 sample dates (2, 4, 8 and 16 weeks) × 2 subsamples. The first experiment was carried out in the dry season (March to June) and the second in the wet season (August to November). The initial quality of the foliage was assessed for each treatment in both experiments. The trees were established as seedlings, were pruned one year later and allowed to regrow. Fresh leaves of the five treatments (selected from 10 trees per treatment) were collected at random for the first experiment when the plants were 7 months old and for the second at 10 months old. Leaf collection for the second experiment was immediately before pruning, and the experiment was conducted in the period following pruning. The leaves for the mixtures were taken from the pairs of the trees in the mixture, and the leaves from one tree were taken from trees growing separately. Polyethylene litterbags of 300 mm × 300 mm with a mesh size of 2 mm were constructed to assess the rate of leaf decomposition. These litter bags were each filled with sufficient fresh leaves to yield the equivalent of about 40 g dry matter. Separate subsamples were taken for dry-matter determination and initial chemical composition, and the actual contents of the bags were as follows:

1. 100 g fresh leaves of *Guazuma*, equivalent to 41 g dry weight
2. 100 g fresh leaves of *Guazuma* + *Leucaena*, equivalent to 40 g dry weight (20 g of *Guazuma* + 20 g of *Leucaena*)
3. 100 g fresh leaves of *Leucaena*, equivalent to 38 g dry weight
4. 118 g fresh leaves of *Moringa* + *Leucaena*, equivalent to 37 g dry weight (19 g *Moringa* + 18 g of *Leucaena*)
5. 136 g fresh leaves of *Moringa*, equivalent to 35 g dry weight

The litter bags were randomly distributed in each experimental block and placed on the soil surface in the alleys between the shrub rows (approximately 50 cm from the trees) on an area previously cleared of leaf litter and other organic material to allow good soil contact with the bags. Once the bags were placed, they were covered with most of the previously displaced litter. Eight litter bags (2 per block) from each treatment were retrieved after field exposure for 2, 4, 8 and 16 weeks. After we collected the litterbags in the laboratory, each litterbag was gently opened and cleaned to remove soil and other detritus. Additionally, leaf remnants' were carefully cleaned with a brush, taking care not to lose any residual vegetative material; then, they were rinsed under running tap water; finally samples were oven-dried at 40–50 °C to constant weight. The weight loss due to

decomposition was calculated from the initial weight and from the final dry weight of the material; once the samples were dried, the weights were recorded, and each sample was ground and analyzed for chemical constituents.

### 2.3. Chemical Analysis of Plant Material Used in the Decomposition Study

Four random subsamples per treatment were taken for litter quality characterization. Total N was determined by the Kjeldahl method [32]. Plant dry matter was digested in concentrated sulfuric acid, whereby organic N was converted to ammonium-N by steam distillation, and the ammonium determined by direct titration with a standard acid. Lignin content was determined following the method proposed by [33]. The polyphenols (Pp) were extracted in hot (80 °C) 70% aqueous acetone and measured colorimetrically using the Folin–Ciocalteu method, and the concentration of condensed tannins (CT) was measured following the butanol-HCl method described by [34].

### 2.4. Statistical Analysis

Data were analyzed as a randomized complete block design at each sampling time. All data were tested for normality using a Kolmogorov–Smirnov test, and non-normal data were transformed when necessary. An ANOVA was carried out on all variates in SPSS using the generalized linear model, and significant differences between means ($p > 0.05$) were analysed by the Tukey test.

Litter decomposition was calculated as the difference between the initial total dry matter and the amount remaining at each retrieval time. The same procedure was carried out for nitrogen. Single and double exponential decay models were fitted to the data, and the second one was found to be a significantly better fit:

$$\mathbf{Y} = \mathbf{A}^{(-\mathbf{k_1 t})} + (1 - \mathbf{A})^{(-\mathbf{k_2 t})} \tag{1}$$

where Y is the percent of the initial free ash mass or N remaining at sampling time t, A is the easily decomposable fraction, and (1-A) is the more recalcitrant fraction. $k_1$ and $k_2$ are the decomposition, or N release, constants for the labile and recalcitrant litter components, respectively. The biological basis of the double exponential decay model is that the litter consists of two fractions, one that decomposes relatively easily (labile fraction) and another more resistant to decomposition (recalcitrant fraction) [35].

### 3. Results

#### 3.1. Climatic Conditions

The weekly climatological conditions (rainfall, maxima and minima temperatures) during the litter bag decomposition experiment are summarized in Figures 1 and 2.

#### 3.2. Chemical Composition of the Shrub Leaves

The chemical composition of the plant material used in the study is presented in Table 2. In the dry period, *Leucaena* leaves had the highest N content (3.3%), while *Moringa* had the lowest (2.1%). The mixtures of leaves had intermediate N content, with similar concentrations for the two mixture treatments (2.8%). Lignin was high in *Guazuma* leaves (18%), while low values were found for *Leucaena* and *Moringa* (both with 10%). There was a pronounced variation in tannin concentration, being high in the *Leucaena* leaves and intermediate in the combination of *Leucaena + Guazuma* (2.3%) and very low in the *Moringa* (0.2%). The ratio of lignin plus polyphenol to N concentration was lowest for *Moringa* leaves (5.8%), and the highest ratio was recorded for *Guazuma* (9.7%). In general, among the treatments, the leaves in mixtures had an intermediate chemical composition in comparison to the single-leaves treatments (Table 2).

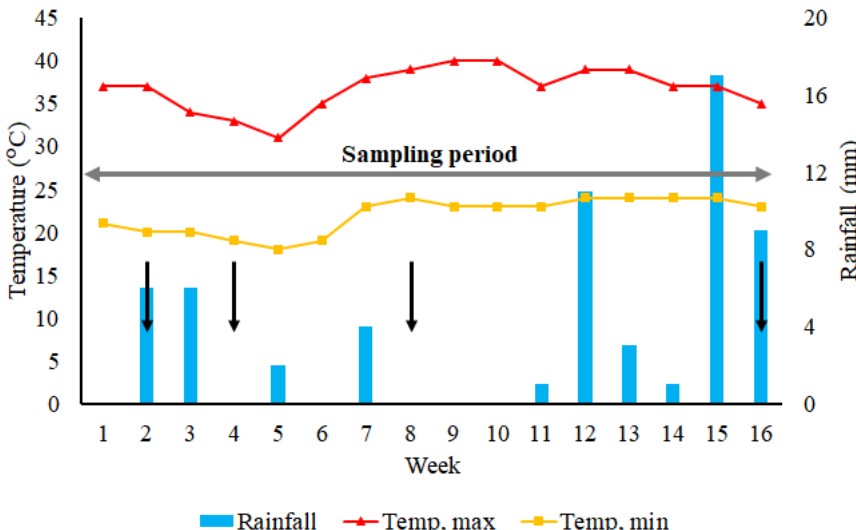

**Figure 1.** Weekly rainfall (mm) and mean weekly maximum and minimum temperature (°C) from March to June (dry period), i.e., till the start of the rainy season. The four sample dates are indicated by arrows.

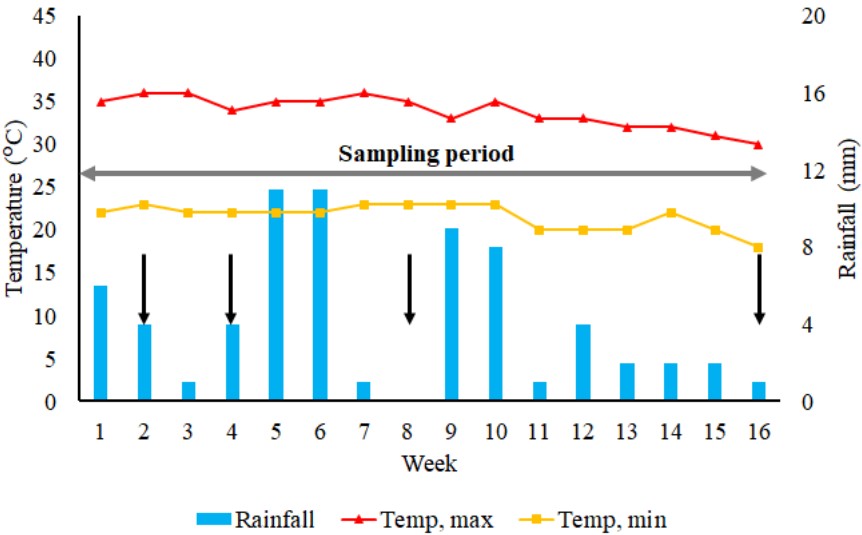

**Figure 2.** Weekly rainfall (mm) and mean weekly maximum and minimum temperatures (°C) from August to November (wet period). The four sample dates are indicated by arrows.

More chemical variability between treatments was found in the rainy period samples. For example, for the N content, values ranged from 1.8 for *Guazuma* to 3.2% for *Leucaena* single leaves. Among leaf mixtures, the range in nitrogen was small (2.4 to 2.9%). Lignin content was similar for all treatments; values ranged from 11 for *Moringa* to 14% for *Leucaena*. However, great differences in tannin concentration were recorded between the treatments, with values ranging from 0.4 to 2.9% for *Moringa* and *Guazuma* respectively, while the mixture formed by *Leucaena* + *Guazuma* had a much higher tannin concentration (2.7%) than the *Leucaena* + *Moringa* mixture (1.5%). *Moringa* leaves were lowest in polyphenol content and had much lower condensed tannin content than *Leucaena* or *Guazuma* leaves. The C/N ratio of *Leucaena* leaves, *Moringa* leaves and the *Leucaena* + *Moringa* leaf mixtures averaged 15:1, whereas the ratio for *Guazuma* was 25:1. The *Leucaena* + *Guazuma* mixture had a C/N ratio closer to *Leucaena* than *Guazuma* (Table 2).

**Table 2.** Chemical composition of leaves from the different species used in the decomposition experiment.

| Species | Content (%) | | | | | Ratio | | | |
|---|---|---|---|---|---|---|---|---|---|
| Period | N | C | Lignin | Pp | Con. Tannin | Pp:N | (L + Pp):N | C:N | CT:N |
| *Dry period* | | | | | | | | | |
| *Guazuma* | 2.3 | 44 | 18 | 4.1 | 1.5 | 1.8 | 9.7 | 19 | 0.68 |
| *Leucaena + G* | 2.8 | 44 | 14 | 4.0 | 2.3 | 1.4 | 6.6 | 16 | 0.82 |
| *Leucaena* | 3.3 | 45 | 10 | 3.9 | 3.1 | 1.2 | 4.2 | 14 | 0.94 |
| *Leucaena + M* | 2.8 | 45 | 14 | 4.0 | 2.3 | 1.4 | 6.4 | 16 | 0.83 |
| *Moringa* | 2.1 | 42 | 10 | 2.2 | 0.2 | 1.0 | 5.8 | 20 | 0.10 |
| *Wet period* | | | | | | | | | |
| *Guazuma* | 1.8 | 44 | 12 | 2.4 | 2.9 | 1.3 | 8.1 | 25 | 1.61 |
| *Leucaena + G* | 2.4 | 44 | 13 | 2.9 | 2.7 | 1.2 | 6.5 | 18 | 1.13 |
| *Leucaena* | 3.2 | 45 | 14 | 3.4 | 2.6 | 1.1 | 5.5 | 14 | 0.83 |
| *Leucaena + M* | 2.9 | 44 | 13 | 2.7 | 1.5 | 0.9 | 5.2 | 15 | 0.53 |
| *Moringa* | 2.6 | 42 | 11 | 1.9 | 0.4 | 0.7 | 4.9 | 16 | 0.16 |

*3.3. Leaf Litter Decomposition*

Changes in the nitrogen content of the residual material in litter bags are shown in Figure 3a,b. A rapid initial release of N in *Moringa* (during the wet period) was observed, with the N content decreasing 64% in the first 2 weeks. On the other hand, *Guazuma* had a slow nitrogen release in the first two weeks, releasing only 9% of its nitrogen. For the dry period, the trend was similar; *Moringa* released the biggest amount of N, whereas *Guazuma* released only 22% of its N. The *Leucaena + Guazuma* mixture had a slow N release during the dry period and a faster release (20%) during the wet period in the first two weeks.

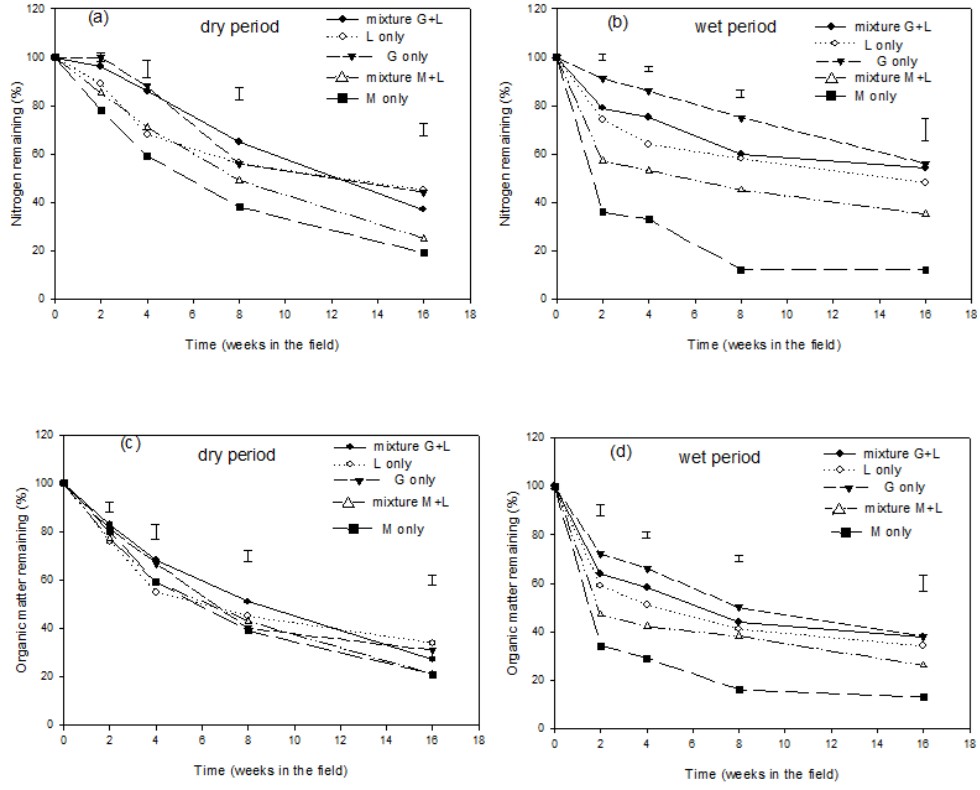

**Figure 3.** Percentage of N ((**a**) and (**b**) during dry and wet season, respectively) and OM ((**c**) and (**d**) during dry and wet season, respectively) remaining in mixtures and pure-residue litter bags from different tree species. Symbol corresponds to the mean values. Vertical bars are standard errors between means (SED).

In the dry period, *Moringa* leaves lost N rapidly, while *Guazuma* showed an initial delay in N release. After 8 weeks of the experiment, the litter of *Moringa* released up to 62% of its initial N. The *Leucaena* and *Guazuma* single leaves released very similar amounts of N (44%) at 8 weeks. On the other hand, the *Leucaena + Moringa* mixture released up to 50% during the first 8 weeks, while *Leucaena + Guazuma* leaves released only 35% of their initial N. Significant differences ($p < 0.001$) were found by the end of the dry-period decomposition study.

In contrast, in the wet period, *Moringa* leaves released up to 64% of the initial N in the first 2 weeks of the leaf decomposition study ($p < 0.001$). The mixture of *Leucaena + Moringa* leaves had an intermediate N release (43%) during the first 2 weeks. After 8 weeks, the nitrogen content remained stable for *Moringa* leaves, followed by the *Leucaena + Guazuma* mixture, with only 6% N release between weeks 8 and 16. *Leucaena* and *Guazuma* leaves had a continuous N release until week 16, with significant ($p < 0.01$) differences between treatments.

*Moringa* had a very fast N release, losing 81% and 88% of its total N after 16 weeks of exposure, in the dry and wet periods, respectively (Figure 3). In contrast, *Guazuma* on its own and the combination of *Leucaena + Guazuma* lost about 46% of the N in the wet period, and *Guazuma* single litter bags and the mixture of *Leucaena + Guazuma* lost 56% and 63%, respectively, in the dry period.

### 3.4. Organic Matter (OM) Decomposition

The organic matter (OM) decomposition pattern of the different litters is shown in Figure 3c,d. In the dry period, no significant differences were found in the decomposition rate between the different treatments. However, *Moringa* leaves lost about 87% of their OM, while *Guazuma* on its own and the *Leucaena + Guazuma* leaf mixture had lost 62% of their OM (Figure 3). In the wet period, significant ($p < 0.01$) differences were found. OM decomposition during the first 2 weeks resulted in a sharp decrease for all treatments but particularly in the *Moringa* and *Leucaena + Moringa* mixture leaves, with losses up to 67%. Losses were 28 and 36% for *Guazuma* and *Leucaena + Guazuma*, respectively (Figure 3c,d). Throughout this period, loss of mass in *Guazuma* leaves proceeded more slowly than for the other treatments, with 56% of the initial leaf mass remaining after 60 days. Decomposition rates of *Moringa* and *Leucaena + Guazuma* mixed leaves decreased after 8 weeks, and decomposition had virtually stopped after 16 weeks. *Guazuma* leaves and the mixture of *Leucaena + Guazuma* had the smallest decomposition, and at the end of the experiment, about 40% of their initial leaf mass had still not decomposed.

### 3.5. Nitrogen Release Patterns

The observed and predicted nitrogen release of individual litter types and litter mixture are presented in Figure 4. Leaf N remaining, expressed as percent of initial N, followed a pattern similar to for OM remaining in the dry season (Figures 4 and 5). During this period, leaves released most of their N during the first 8 to 16 weeks of decomposition, *Moringa* leaves and the *Leucaena + Moringa* leaf mixture ranked first, in terms of N release rate, having lost 81 and 75% of initial N, respectively after 16 weeks. During the first two weeks, N release was slowest from *Guazuma* leaves and from the mixed *Leucaena + Guazuma* leaves; these two materials mineralized their leaf N at considerably slower rates between 2 and 8 weeks (Figure 4a,b). At 16 weeks, percent N release was greatest for *Moringa* and the *Leucaena + Moringa* mixture. On the other hand, *Guazuma* and *Leucaena* leaves still retained about 45% of their leaf N at the end of the study period.

In the wet season, nitrogen was released from all species during the first two weeks. Differences in leaf decomposition were significant between the species (Table 3). Over the first 2 weeks incubation, a rapid N release was observed for *Moringa* leaves and the *Leucaena + Moringa* mixture, with more than 60% of the N in the *Moringa* leaf litter being released during this period (Figure 3). However, between weeks two and four, the rate of N release slowed down generally for all species but particularly for *Moringa*, probably due to conditions being dry during this period. The release of N from *Guazuma* leaves and the

*Leucaena + Guazuma* mixture was especially slow, with 56% and 54%, respectively, remaining after 16 weeks. *Leucaena* had an intermediate rate of N release, with 48% remaining at the end of the experimental period. This compares with less than 15% of N remaining after weeks in the *Moringa* leaves (Figure 3).

In general, the pattern of N released from all litter types followed two phases. A fast N release was followed by a slower rate of loss or no change towards the end of the study. Differences in N release were apparent within the mixed-species litter combinations. The initial leaf litter from the *Leucaena + Moringa* mixture released about 55% of the N content in the first two weeks, whereas the *Leucaena + Guazuma* mixtures only released about 20% of the leaf N over the same period. On the other hand, *Leucaena* leaf had an intermediate rate of N release. More than half of the N in the *Leucaena + Moringa* mixtures was released during the first 8 weeks, but *Leucaena* leaves on their own and the *Leucaena + Guazuma* mixture released about 40% of their N by the end of the experimental period.

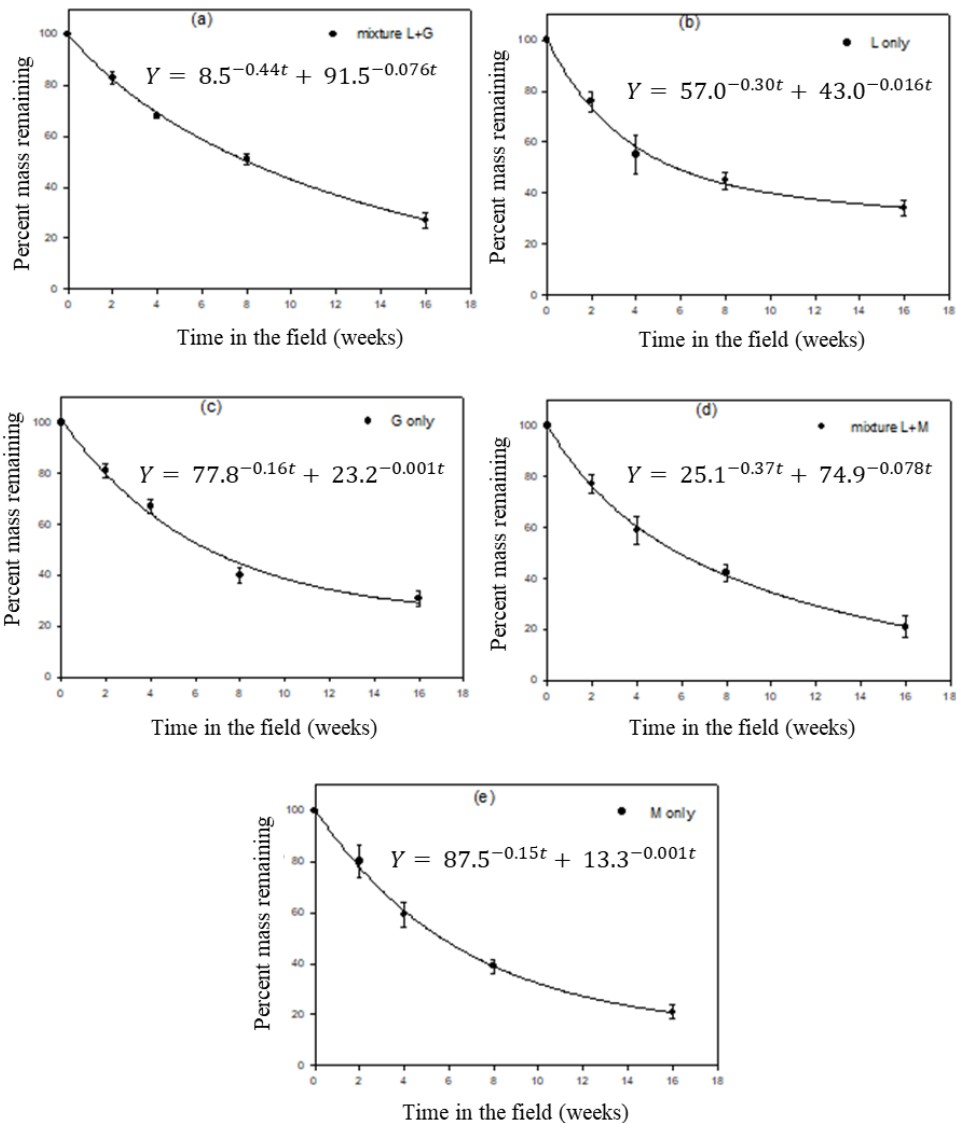

**Figure 4.** Modelled and measured mass remaining from decomposing leaves ((**a**) = *Leucaena + Guazuma*, (**b**) = *Leucaena*, (**c**) = *Guazuma*, (**d**) = *Leucaena + Moringa*, (**e**) = *Moringa*) for the dry period. Bars are SE.

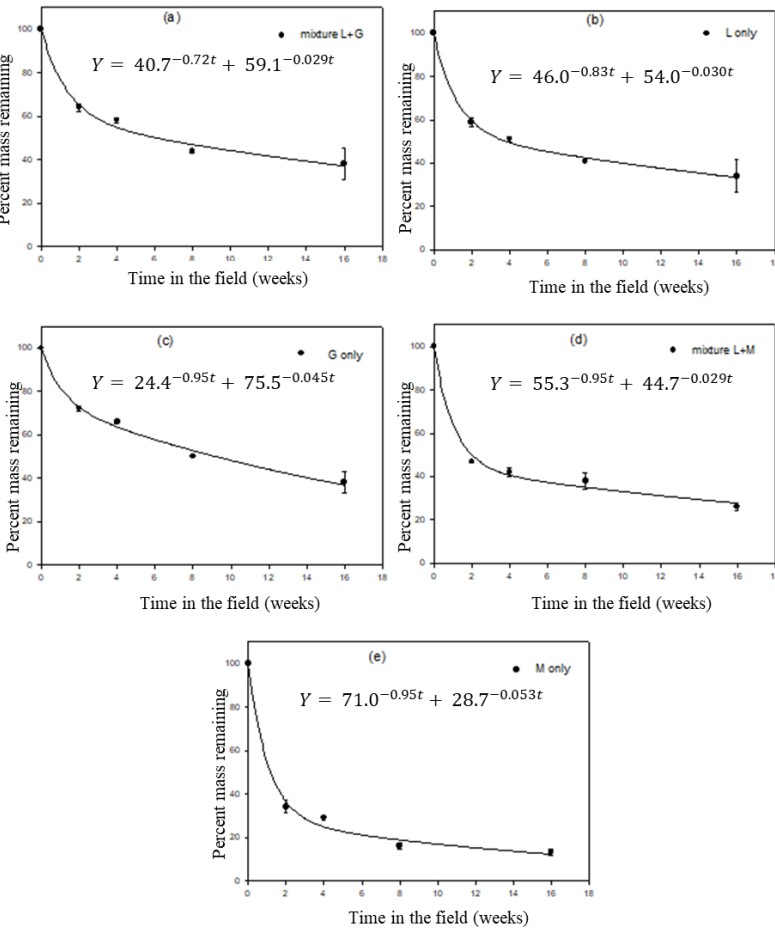

**Figure 5.** Modeled and measured mass remaining from decomposing leaves ((**a**) = *Leucaena* + *Guazuma*, (**b**) =*Leucaena*, (**c**) = *Guazuma*, (**d**) = *Leucaena* + *Moringa*, (**e**) = *Moringa*) for the wet period. Bars are SE.

**Table 3.** Correlation coefficients (r) among decomposition parameters and leaf chemical quality for the dry and wet periods.

| Decomp. Parameter | %N | % Lignin (L) | Pp | CT | Pp/N | (L + Pp)/N | C/N | CT/N | L/N |
|---|---|---|---|---|---|---|---|---|---|
| | | | | Dry period | | | | | |
| $K_L$ | 0.68 | 0.01 | 0.55 | 0.70 | 0.01 | −0.33 | −0.75 | 0.68 | −0.39 |
| $K_R$ | 0.42 | 0.14 | 0.46 | 0.48 | 0.08 | −0.13 | −0.51 | 0.51 | −0.18 |
| $C_L/C_R$ | −0.78 | −0.25 | −0.85 | −0.89 | −0.34 | 0.12 | 0.83 | −0.93 * | 0.20 |
| K | 0.64 | −0.51 | 0.10 | 0.50 | −0.33 | −0.63 | −0.58 | 0.35 | −0.65 |
| | | | | Wet period | | | | | |
| $K_L$ | −0.15 | −0.55 | −0.62 | −0.52 | −0.44 | −0.08 | 0.19 | −0.27 | −0.02 |
| $K_R$ | −0.48 | −0.92 * | −0.87 * | −0.51 | −0.37 | 0.09 | 0.39 | −0.18 | 0.15 |
| $C_L/C_R$ | 0.34 | −0.58 | −0.60 | 0.97 ** | −0.97 ** | −0.78 | −0.52 | −0.92 * | −0.73 |
| K | 0.46 | −0.47 | −0.52 | −0.98 ** | −0.99 *** | −0.84 | −0.60 | −0.96 ** | −0.80 |

Note: *, ** and *** are significant at 5% y 1%, 0.1% levels, respectively.

Patterns of leaf decomposition were best characterized by the double exponential model, although the small number of samplings affected the significance of the regression values (Figure 4). The k values were consistently higher, particularly for *Moringa* and *Leucaena* material in comparison to the combination of the *Leucaena* + *Guazuma* mixture leaves. Decomposition rates showed more variation during the dry period (0.15 to 0.44 wk$^{-1}$) than during the rainy period (0.72 to 0.95 wk$^{-1}$). This was probably due to the greater relative variability in moisture conditions in the litter in the dry period.

Overall, *Moringa* and *Leucaena + Moringa* material disappeared faster than the other materials. The proportions of recalcitrant material were found to range from 91% (*Leucaena + Guazuma*) to only 13% (*Moringa*) in the dry period (Table 3). The decomposition constants (k) for *Moringa* and *Leucaena + Moringa* were estimated to be 0.15 and 0.37 wk$^{-1}$, respectively. On the other hand, *Leucaena* had an intermediate proportion of recalcitrant material (43%), and the decomposition constant (k) was about 0.30 wk$^{-1}$ for the same period (Table 3 and Figure 4).

### 3.6. Relationship between Organic Matter Decomposition and Chemical Composition

The decomposition constants and N release were estimated from the models. No significant correlations were found between the initial chemical component of the leaf material and the organic matter remaining at 16 weeks during the dry period (Table 3). On the other hand, significant correlations were found for the wet period. This emphasizes the role of not only the chemical composition of the leaves litter on its decomposition but also the role of the climatic conditions in both periods (Table 3).

### 3.7. Relationship between Nitrogen Release and Leaf Chemical Composition

With respect to N release, only the ratios of condensed tannin (CT) and polyphenols (Pp) to N were significantly correlated with nitrogen released from litter bags in the field after 16 weeks (Table 4). CT content showed a high correlation coefficient ($r^2 = 99$; $p > 0.001$). The significant correlation found with the k constant emphasized the role of the chemical composition of a specific material on its rate of decomposition.

**Table 4.** Correlation coefficients (r) among N release and leaf chemical quality for the dry and rainy periods.

| Rate Constants | %N | % Lignin (L) | Pp | CT | Pp/N | (L + Pp)/N | C/N | CT/N | L/N |
|---|---|---|---|---|---|---|---|---|---|
| | | | | Dry period | | | | | |
| $Kn_L$ | 0.22 | −0.86 * | −0.54 | −0.04 | −0.76 | −0.75 | −0.15 | −0.27 | −0.72 |
| $Kn_R$ | 0.18 | 0.15 | 0.27 | 0.23 | 0.08 | −0.04 | −0.25 | 0.28 | −0.06 |
| $N_L/N_R$ | −0.86 * | 0.16 | −0.58 | −0.85 * | 0.08 | 0.50 | 0.91 * | −0.78 | 0.57 |
| Kn | 0.13 | −0.01 | 0.11 | 0.14 | −0.07 | −0.14 | −0.19 | 0.02 | 0.15 |
| | | | | Wet period | | | | | |
| $Kn_L$ | −0.11 | −0.51 | −0.58 | −0.50 | −0.44 | −0.09 | 0.16 | −0.027 | −0.04 |
| $Kn_R$ | 0.01 | −0.79 | −0.77 | −0.86 * | −0.79 | −0.48 | −0.19 | −0.69 | −0.43 |
| $N_L/N_R$ | 0.33 | −0.57 | −0.59 | −0.95 ** | −0.94 ** | −0.77 | −0.53 | −0.91 * | −0.73 |
| Kn | 0.33 | −0.60 | −0.63 | −0.99 *** | −0.98 ** | −0.76 | −0.49 | −0.92 * | −0.71 |

Note: *, ** and *** are significant at 5% y 1%, 0.1% levels, respectively.

Figure 5 shows the decomposition pattern for the rainy period. *Leucana + Guazuma* (5a) mixture had a higher decomposition rate, compared to *Guazuma* alone. *Leucaena* (5b) decomposed faster over the first 8 weeks, while *Guazuma* had the slowest rate of decomposition (5c). A rapid loss of leaf material in litterbags was observed with mixed *Leucaena + Moringa* leaves and *Moringa* leaves during the initial decomposition stage (Figure 5d,e). The rate of the mass loss of *Moringa* leaves was large during the first two weeks. In this period, the recalcitrant materials were 76% and 29% for pure *Guazuma* and pure *Moringa*, respectively.

*Moringa* leaves released more N during the first 8 weeks of decomposition than those of the other species. The observed decay rates of single and mixed-species leaf combinations of *Guazuma* and *Leucaena + Guazuma* retained more N than pure *Leucaena* litter bags. *Guazuma* leaves and the combination of *Leucaena + Guazuma* leaves released most of their N during the 4 to 8 weeks of decomposition. Between 2 and 4 weeks, N immobilization apparently occurred in *Guazuma*, which also had a poor fit to the dataset, and a similar trend was also observed in the mixed combination of *Leucaena + Guazuma* (Figure 6a,c).

The mixture Leucaena+Moringa and Leucaena, Moringa alone exhibited similar decomposition patterns that were significant different from that of Guazuma (Figure 7). All three species had a rapid phase of decomposition lasting about one month followed by a slower phase. In Moringa the rapid phase was more pronounced (Figure 7e).

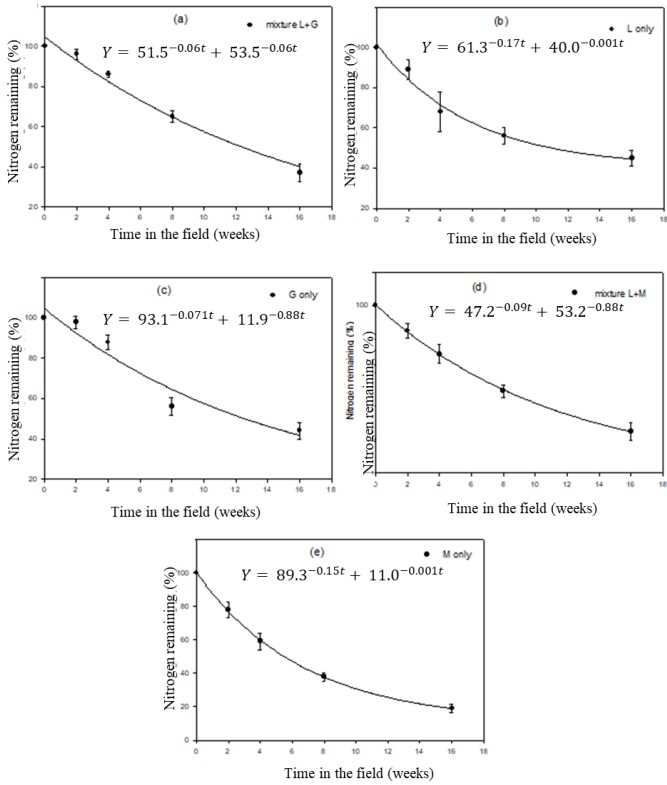

**Figure 6.** Modeled and measured nitrogen from decomposing leaves ((**a**) = *Leucaena + Guazuma*, (**b**) = *Leucaena*, (**c**) = *Guazuma*, (**d**) = *Leucaena + Moringa*, (**e**) = *Moringa*) for the dry period. Bars are SE.

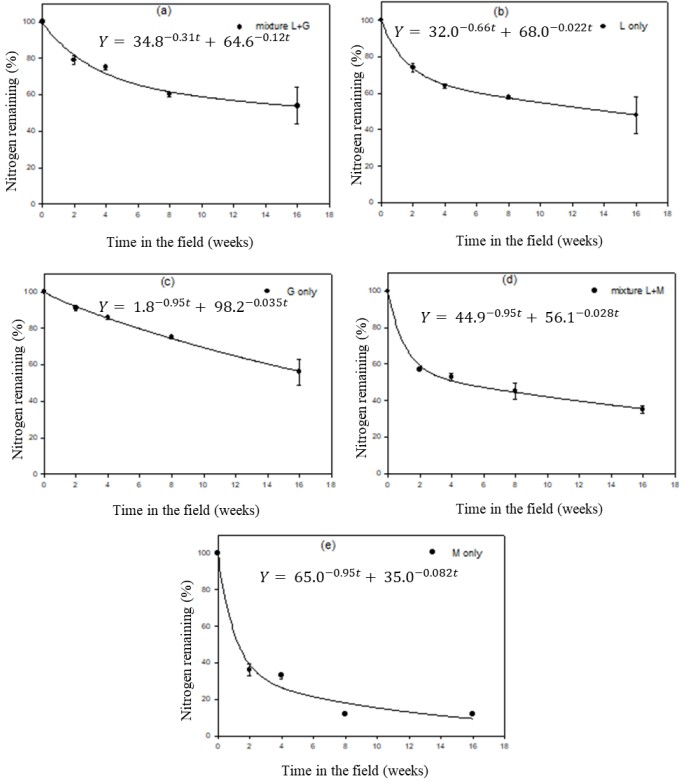

**Figure 7.** Modeled and measured nitrogen from decomposing leaves ((**a**) = *Leucaena + Guazuma*, (**b**) = *Leucaena*, (**c**) = *Guazuma*, (**d**) = *Leucaena + Moringa*, (**e**) = *Moringa*) for the wet period. Bars are SE.

## 4. Discussion

### 4.1. Chemical Composition of Leaves in the Study

The leaf residues of the species under study had different qualities. Those from the non-leguminous shrubs contained less N (Table 5). The soluble polyphenol content of the *Guazuma* leaves was high in comparison to that from *Moringa* but similar to the *Leucaena*. Tree pruning (only leaves) contributed 919 kg to 5653 kg of dry matter and 19 to 158 kg of N per hectare (Table 5). *Moringa* leaves and the mixed materials of *Leucaena* with *Moringa* had less N content in the leaves than *Leucaena* and *Leucaena + Guazuma* mixtures. It has been reported that the contribution of different types of litter contributes to the release and availability of nutrients [36].

**Table 5.** Nitrogen concentration and N (kg ha$^{-1}$) soil inputs from decomposed leaves in shrubs.

| Specie | Initial | Pruning Leaves | Initial N | %N and kg N Released ha$^{-1}$ (Time in Weeks) | | | | | | | |
| | | | | 2 | | 4 | | 8 | | 16 | |
| Dry Period | %N | kg ha$^{-1}$ | kg ha$^{-1}$ | % | kg | % | kg | % | kg | % | kg |
| --- | --- | --- | --- | --- | --- | --- | --- | --- | --- | --- | --- |
| *Moringa* | 2.1 | 919 | 19 | 22 | 4 | 41 | 8 | 62 | 12 | 81 | 15 |
| Leuc (M) | 2.8 | 3914 | 62 | 15 | 9 | 29 | 18 | 51 | 32 | 75 | 47 |
| *Guazuma* | 2.3 | 2681 | 110 | 0 | 0 | 12 | 13 | 44 | 48 | 56 | 62 |
| *Leucaena* | 3.3 | 4559 | 150 | 11 | 17 | 32 | 48 | 44 | 66 | 55 | 83 |
| Leuc (G) | 2.8 | 5653 | 158 | 4 | 6 | 14 | 22 | 35 | 55 | 63 | 100 |
| **Wet period** | | | | | | | | | | | |
| *Moringa* | 2.6 | 919 | 24 | 64 | 15 | 67 | 16 | 88 | 21 | 88 | 21 |
| *Guazuma* | 1.8 | 2681 | 48 | 11 | 5 | 14 | 7 | 25 | 12 | 44 | 21 |
| Leuc (M) | 2.9 | 3914 | 114 | 43 | 49 | 47 | 54 | 55 | 63 | 65 | 74 |
| Leuc (G) | 2.4 | 5653 | 136 | 21 | 29 | 25 | 34 | 40 | 54 | 46 | 63 |
| *Leucaena* | 3.2 | 4559 | 146 | 26 | 38 | 36 | 53 | 42 | 61 | 52 | 76 |

Nitrogen release from *Leucaena* residue agreed with the results of a study conducted at the International Institute of Tropical Agriculture [16] under alley cropping conditions, in which 40% of the N was released from *Leucaena* residue within a period of 4 weeks, and with a study conducted in Yucatan with leaves of native trees and shrub species under silvopastoral systems [37].

In the wet period, *Guazuma* also showed clearly that it can be used alongside other shrubs for fodder or as an N contribution in intercropping systems. With the exception of *Guazuma* in the wet period, the N lost during the 16 weeks decomposing was over 55%, which could be considered high enough to justify the use of these species to supply high-quality foliage for use both as fodder and to supply N directly to the soil via fresh leaves of litterfall. It is important to consider here that this amount of N does not include the edible stems, which also had a large N concentration and can supply an important amount of available nitrogen to the system [38].

The combination treatments resulted in very similar N contents (2.8%) for the dry period, while for the rainy season, the combination of Leucana + *Guazuma* had 2.4% and *Leucaena + Moringa* had 2.9%. Mixing *Leucaena + Guazuma* and *Leucaena + Moringa* leaves also resulted in high polyphenol content. According to the litter characterization of [34], plant materials containing at least 2.5% N, less than 15% of lignin and less than 4% of polyphenol are usually described as being of high quality. Our results fall within these parameters except for *Guazuma*, which is recognized as a shrub of low quality that is likely to temporarily immobilize N during decomposition [39]. However, mixtures of contrasting quality would offer important scope for synchronizing the nutrient release in crops or intercropping systems [40]; this is because there is an intrinsic variability in the loss of mass and the release of nutrients from one species to another; however, with fodder mixtures, this variability can be reduced [39]. This temporal variation in the dynamics of litter decomposition and nutrient release is particularly of interest in regions with marked seasonality (wet and dry periods) such as the study area; other studies have found a seasonal pattern in the litter decomposition associated with variations in environmental conditions related to *Eucalyptus grandis* trees [41] and also has reported seasonal variation in leaf litter amount production [42].

### 4.2. Decomposition Rates and Quality of Shrub Leaves

Comparison of results showed considerable differences in decomposition rates between the five materials studied. The most rapid decomposition was recorded for *Moringa* leaves and the slowest for *Guazuma* leaves. The smaller amount of N released and the dry-matter loss in *Guazuma* leaf litter were probably due to the temporarily immobilization of N during decomposition, as their leaves had less than 2.3% N. This is considered to be at the low end of the range for the release of N [33]; leaf litter quality influences mass loss and nutrient dynamics, while litter diversity can affect the rate of these processes [43].

The contents of N in the shrub species studied herein were all above this critical concentration and almost all had C:N ratios below 20, except for *Guazuma* in the wet period, which had a C:N ratio of 25. Leaf litter with high C:N can also provide K and other nutrients, which contributes to improving soil fertility [44]. However, results from this experiment only partly matched the prediction of the earlier work cited above; for example, the proportion of N released from the combination of *Leucaena* + *Guazuma* leaves was much less than that released from the combination of *Leucaena* + *Moringa* in both wet and dry seasons, although the leaves had similar C:N ratios.

These differences could be due to the large tannin content and the (lignin + polyphenol)-to-N ratio in the mixture formed by leaves of *Leucaena* + *Guazuma*, as polyphenols are compounds capable of binding plant proteins. Furthermore, they inhibit nitrification, as well as decomposition and nutrient cycling [45]. Condensed tannins (CTs), also called proanthocyanidins, are mixtures of polymers with different degrees of polymerization (commonly found in trees or shrub species), decompose more slowly and are also considered to have an important impact on nitrogen immobilization in soils, particularly when fresh litter materials are in close contact with the soil [46,47]. The incorporation of legume species in the production systems, intercropping them with other species, increases the amount of available nitrogen in the soil through the fixation of atmospheric nitrogen and, on the other hand, decreases the loss of nitrogen by infiltration, nitrification and denitrification [48].

The ratio of polyphenol to N was the parameter that could best be used to predict the N mineralization of tropical legumes, with a critical value of 0.5 [49]. While the lignin:N ratio has been reported to not always predict N mineralization by itself, a negative correlation has been found between nitrogen mineralization and the lignin:N ratio [50]. The decomposition patterns of the leaves studied herein were related to their polyphenol-to-N ratios. However, the lignin + polyphenol-to-N ratio was the parameter that best predicted the N mineralization of the leaves of five tree species in a hedgerow intercropping experiment [51].

The pattern of steadily declining decomposition rates of all treatments in both periods suggests that first the soluble and easily degraded compounds are utilized, with the remaining biomass being more resistant to decomposition. Over an 8-week period, between 50 and 60% of the leaf mass was lost in the dry season, compared with 60 and 80% in the wet period. Similar trends of decomposition rates in tropical conditions have been reported [37,52], using leaves from leguminous and non-leguminous species. Regarding N release, the three species studied exhibited contrasting behaviors. *Moringa* leaves released much of their N almost instantly; N release from *Leucaena* leaves was intermediate, while *Guazuma* leaves released their N still more slowly. No data were found in the literature for *Moringa* and *Guazuma* to make a comparison of their decomposition behavior.

*Leucaena* leaves had the highest N concentration and the lowest C/N ratio, but because of their high polyphenol content, they decompose slightly more slowly than those of *Moringa*. An additional factor that may influence the rate of decomposition is the micro-climate, e.g., the alley formed in the *Moringa* and *Leucaena* (M) plots had taller trees with a closed canopy that reduced soil water evaporation and created better conditions for the decomposition process. Soil macrofauna populations were not assessed in this study. However, more humid micro-climate conditions created in the *Leucaena* and *Moringa* plots would lead to higher populations of these organisms, many of which are decomposers [53]. Furthermore, a bigger macrofaunal population in decomposing litter from hedgerows formed by *L. leucocephala* and *C. calothyrsus* than in litter from single-species stands [54].

They attributed these differences to more suitable moisture, temperature and soil conditions that can enhance faunal populations. In this project, the *Guazuma* plants were mostly small, and in some cases, they did not fully cover the soil area where the litter bags were placed. Furthermore, the initial N content in *Guazuma* leaves could be below the requirements for decomposers.

## 5. Conclusions

The three species under study demonstrated their ability to produce considerable dry matter and accumulate significant amounts of N in short time periods. The initial mass loss from the litter was high and rapid in the rainy period in comparison to the dry period. The residue disappearance pattern of *Moringa*, *Leucaena* and *Leucaena + Moringa* followed an asymptotic model, with more than 80% of the original residue released during the 16-week study period. *Moringa* residue decomposed significantly faster than all other treatments, followed closely by the combination of *Leucaena + Moringa*. *Leucaena* leaves released their N at intermediate rates, but N in *Guazuma* and the combination of *Leucaena + Guazuma* mineralized considerably more slowly.

Polyphenols appear to be the most important factor influencing the rates of decomposition as they bind to N in the leaves, forming compounds resistant to decomposition. The lignin and CT contents of *Moringa* leaves were lower than other two species, and although numerous studies have suggested that the initial lignin content is a reasonable predictor of the rate of decomposition, our data suggest that lignin content is a poor index on its own.

**Author Contributions:** Conceptualization, F.J.S.-S.; data curation, M.d.C.T.-G.; formal analysis, L.R.y.A.; investigation, M.d.C.T.-G., O.O.Á.-R., L.R.y.A. and F.J.S.-S.; methodology, L.R.y.A. and F.J.S.-S.; project administration, F.J.S.-S.; supervision, F.J.S.-S.; visualization, M.d.C.T.-G., O.O.Á.-R., L.R.y.A. and F.J.S.-S.; writing—original draft, M.d.C.T.-G., O.O.Á.-R. and L.R.y.A.; writing—review and editing, M.d.C.T.-G., O.O.Á.-R., L.R.y.A. and F.J.S.-S. All authors have read and agreed to the published version of the manuscript.

**Funding:** This research received no external funding.

**Institutional Review Board Statement:** Not applicable.

**Data Availability Statement:** For additional information contact the author by correspondence.

**Acknowledgments:** We thank R. Carcaño and E. Euan for help in sampling collection and processing.

**Conflicts of Interest:** The authors declare no conflict of interest.

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
