# Peer review of "Decomposition and Nitrogen Release Rates of Foliar Litter from Single and Mixed Agroforestry Species under Field Conditions"

_agriculture, doi:10.3390/agriculture13010222_

Round 1
Reviewer 1 Report
Thanks for invitation to review the manuscript “Decomposition and nitrogen release rates of foliar litter from single and mixed agroforestry species under field conditions” The present manuscript is suitable for the journal and its could consider after further modification the suggestion are as under.
## The section intro is well written need to update with curret situation. the situation is current or old? In the tropics and subtropics, millions of people have no food security; about 60% of rural communities are permanently affected by a decline in household food production, with sub-Saharan Africa, Central Asia, the Caribbean, and parts of Latin America includ- ing Mexico suffering most
##Table 1. Physical and chemical characteristics of the experimental area (mean values by block)
Put the suitable reference of the analytical analysis.
|
Block |
Stone |
pH |
N |
C |
C:N |
P |
Exch K |
Exch Ca |
Exch Mg |
|
% |
% |
ratio |
mg kg-1 |
||||||
|
I |
78 |
7.8 |
0.89 |
6.4 |
7.2 |
28 |
530 |
872 |
352 |
|
II |
60 |
7.8 |
0.98 |
5.0 |
5.1 |
45 |
565 |
824 |
328 |
|
III |
79 |
7.9 |
0.99 |
7.2 |
7.3 |
81 |
457 |
1077 |
310 |
|
IV |
79 |
7.9 |
0.96 |
6.1 |
6.4 |
111 |
517 |
1573 |
388 |
|
Mean |
74 |
7.8 |
0.95 |
6.2 |
6.5 |
66 |
517 |
1086 |
345 |
## make it colorful for batter representation. Figure 1. Weekly rainfall (mm) and mean weekly maximum and minimum temperature (ºC) from March to June (dry period), i.e. till the start of the rainy season. The four sample dates are indicated
by arrows.
## Change the sentence The decomposition constants and N release were estimated directly from the models. No significant correlations were found between the initial chemical component of the leaf material and the organic matter remaining at 16 weeks in field litter bags incubated during the dry period (Table 3). However, significant correlations were found for the wet period. This emphasizes the role of not only the chemical composition of the specific material on its decomposition but also the role of the climatic conditions, as we can see from Table 3. In the dry period the decomposition rates were governed more by the climatic conditions than by the chemical characteristics of the material used in the study.
## Change the sentence The N release was controlled more by polyphenol and CT than by lignin or N content. Polyphenols appear to be the most important factor influencing rates of decomposition as they bind to N in the leaves, forming compounds resistant to decomposition. The lignin and CT contents of Moringa leaves were lower than those of both the other species, and although numerous workers have suggested that the initial lignin content is a reasonable
Rest things are ok
Thanks much
Reviewer 2 Report
All results presented must be improved throughout, the figures were not prepaered well, so that they would give readers some incorrect information. Also, the data statistics is not shown as scientific way, must be improved. The manuscript is narrative in form. It is difficult to read.
Reviewer 3 Report
The article is of interest for practical use in agricultural production, especially on degraded soils. The use of leaf litter or plant parts in the tropics to improve soil fertility is an important factor in increasing crop yields and ensuring food security of the population. The authors are presented with a number of questions on the methodology of conducting experiments and suggestions for improving the presentation of research results.
1. Why is block 1 in bold type in table 1? How is it different from other blocks?
2. Why were fresh leaves cut off in the experiment, and fallen ones were not collected? The composition of trace elements in fresh and fallen leaves may differ slightly, which will eventually lead to different results of nitrogen accumulation in the soil. The mechanisms and rate of decomposition of fresh and fallen leaves may also differ.
3. Why were the leaves dried in litterbags and not on the soil? It is necessary to simulate the conditions of leaf decomposition in the natural environment. Moisture could linger in the litterbags, which could affect the results of the experiment.
4. How was the control of cleaning the litterbags from the soil, visually? There could be a violation of the identity of the experimental conditions for all bags.
5. In the end, how important is the rate of nitrogen formation as a result of decomposition for agriculture? All nitrogen in both 2 and 16 weeks will be in the soil layer. Thus, the rate of decomposition is not the main factor here, but the amount of nitrogen formed is a more important result of the study.
Round 2
Reviewer 1 Report
Dear Sir,
The present Paper can be accepted now.
Thanks
Reviewer 2 Report
I think the article is corrected and can be accepted. I only suggest English editing.
Reviewer 3 Report
The authors have corrected all comments. Clarifications were made on the methodology of the experiment. Literary references corrected. According to the reviewer, the questions for the first version of the article will help the authors to plan future experiments more correctly. The article after corrections has become much better and can be recommended for publication.